# SPATIO-TEMPORAL GRAPH UNLEARNING

## ABSTRACT

Spatio-temporal graphs are widely used in modeling complex dynamic processes such as traffic forecasting, molecular dynamics, and healthcare monitoring. Recently, stringent privacy regulations such as GDPR and CCPA have introduced significant new challenges for existing spatio-temporal graph models, requiring complete unlearning of unauthorized data. Since each node in a spatio-temporal graph diffuses information globally across both spatial and temporal dimensions, existing unlearning methods primarily designed for static graphs and localized data removal cannot efficiently erase a single node without incurring costs nearly equivalent to full model retraining. Therefore, an effective approach for complete spatio-temporal graph unlearning is a pressing need. To address this, we propose CallosumNet, a divide-and-conquer spatio-temporal graph unlearning framework inspired by the corpus callosum structure that facilitates communication between the brain's two hemispheres. CallosumNet incorporates two novel techniques: (1) Enhanced Subgraph Construction (ESC), which adaptively constructs multiple localized subgraphs based on several factors, including biologically-inspired virtual ganglions; and (2) Global Ganglion Bridging (GGB), which reconstructs global spatio-temporal dependencies from these localized subgraphs, effectively restoring the full graph representation. Empirical results on four diverse real-world datasets show that CallosumNet achieves complete unlearning with only 1% - 2% relative MAE loss compared to the gold model, significantly outperforming state-of-the-art baselines. Ablation studies verify the effectiveness of both proposed techniques.

## 1 INTRODUCTION

Recent advanced spatio-temporal graph models effectively capture complex dynamic processes, such as urban traffic flows, molecular interactions, and healthcare monitoring, by harnessing both spatial adjacency and temporal continuity. However, the broad deployment of these powerful models increasingly faces stringent privacy regulations, such as the General Data Protection Regulation (GDPR)European Union (2016) and the California Consumer Privacy Act (CCPA)California State Legislature (2018), which necessitate the complete removal or *unlearning* of sensitive user data upon request. As a result, ensuring compliance with these privacy requirements often requires retraining the entire spatio-temporal graph model to preserve privacy for individual nodes, a process that, while essential, introduces additional computational demands.

**Motivating scenario.** Taking a mobile–location service (e.g., Google Maps) as an example, Figure 1(a) shows smartphones (nodes) forming a richly coupled spatio-temporal graph stream of time-stamped GPS signals. Suppose a subset of users revokes consent for their location data, necessitating the deletion of these devices and all incident edges, as shown in Figure 1(b). Simply dropping the raw records (Figure 1(c)) does not fully satisfy the deletion requirement, as it fails to eliminate the latent influence of the revoked users. Conversely, retraining the entire model from scratch after purging those records (Figure 1(d)) erases the influence but fragments long-range spatial and temporal paths, severely degrading accuracy and interpretability for the remaining users, with a prohibitively high retraining cost.

In such scenarios, it is desirable to have an unlearning method capable of undoing the impact of individual graph nodes both spatially and temporally. However, existing unlearning pipelines fail when applied to spatio-temporal (ST) graphs. In static graphs, removing a vertex typically only perturbs a small neighborhood, meaning partition-retrain or lightweight fine-tuning is often sufficient.

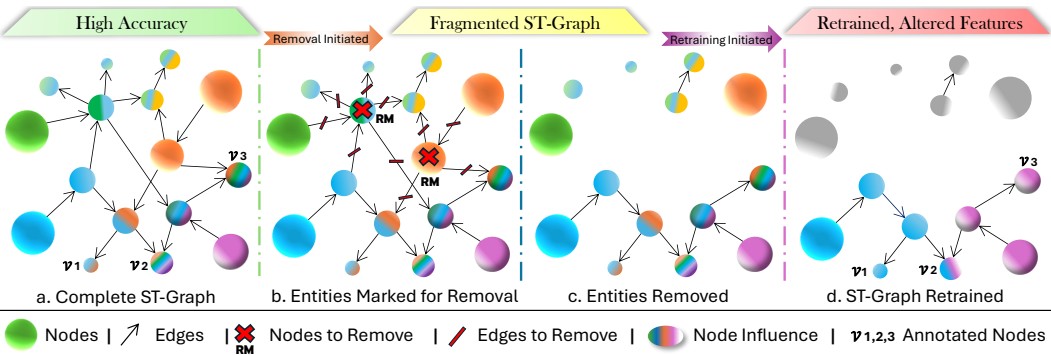

Figure 1: Unlearning on a spatio-temporal graph. (a) A fully connected ST-Graph yields high accuracy; node size encodes impact, color encodes evolving features, and arrows denote spatio-temporal edges. (b) Red marks indicate users who revoke data-use consent; their nodes and incident edges must be erased. (c) Deleting raw records satisfies compliance yet leaves residual influence (faded arrows) inside the model. (d) Retraining after deletion purges influence but fragments the graph and distorts remaining node features ($v_1, v_2, v_3$), harming accuracy.

In contrast, ST graphs are fundamentally different: messages propagate across both space and time, meaning a single node can influence the entire history of the graph. This presents a key challenge: achieving 100% unlearning requires computation nearly equivalent to retraining the model from scratch. Classic data-sharding methods, while useful, risk severing critical spatial or temporal connections, thereby damaging the global spatio-temporal dependencies. Additionally, some methods aim to reduce node influence, yet fail to meet the requirement of 100% unlearning. Consequently, the problem remains unsolved.

In this study, inspired by the structure of the corpus callosum (see Figure 2), we propose CallosumNet. The corpus callosum, connecting the left and right hemispheres of the brain, allows each hemisphere to focus on its respective tasks while sharing information and collaborating. Similarly, CallosumNet applies a divide-and-conquer approach: it builds locally enhanced subgraphs and compensates for the global context through a lightweight meta-graph integration layer to support unlearning in spatio-temporal prediction tasks.

**Challenge 1: How can CallosumNet apply a divide-and-conquer approach without breaking spatio-temporal dependencies, which would lead to a degradation of the model's spatio-temporal prediction capability?**

**Solution 1:** Straightforward cuts can break high-order dependencies, thereby eroding predictive quality. Two novel techniques introduced by CallosumNet are **Enhanced Subgraph Construction (ESC)** and **Global Ganglion Bridging (GGB)**. ESC focuses on constructing well-defined local sub-graph models that enhance the ability to capture regional spatio-temporal attributes, while GGB, building on ESC, establishes a lightweight global integration slot (a meta-graph layer) that fuses information across sub-graphs.

**Challenge 2: How does CallosumNet ensure 100% unlearning?**

**Solution 2:** In Step 1, CallosumNet constructs multiple enhanced spatio-temporal sub-graphs, each of which is closed, with node influence restricted to the respective sub-graph, preventing any spillover effects to other sub-graphs. In Step 2, the weights of all sub-graphs are frozen and remain unaffected. The **Global Ganglion Bridging**, containing global information, rapidly resets and clears after each unlearning process, ensuring that 100% unlearning is achieved.

**Contributions.** We reveal the limitations of current unlearning approaches in ST graphs and propose a divide-and-conquer solution: carving the ST-graph into coherence-preserving local sub-graphs and recovering global context via a lightweight integration layer. CallosumNet implements this approach, combining ESC for local sub-graph construction and GGB for global integration. Across four real-world benchmarks, CallosumNet achieves 100% exact unlearning with only 1%–2% relative MAE loss compared to the gold model.

## 2 RELATED WORK

**Unlearning Methods.** Most existing methods target static graphs. **SISA** Bourtoule et al. (2021) randomly shards the training set and trains each shard in isolation; when naively applied to graphs—especially spatio-temporal ones—such random sharding severs structural and temporal links, so temporal coherence cannot be preserved. **STEPs** Guo et al. (2025) follows the same idea but, for ST graphs, simply strings together broken mini-graphs (or orphan nodes) without reconstructing the lost links, leaving temporal paths fragmented. **GraphEraser** Chen et al. (2022) adopts property-aware sharding to preserve graph structure and retrains only the affected sub-GNNs, but it is evaluated solely on static snapshots and cannot address global spatio-temporal entanglement. **GraphRevoker** Zhang et al. (2025) improves shard-level retraining with property-aware splits and contrastive aggregation, but it too is validated only on static graphs and therefore leaves cross-time dependencies unresolved.

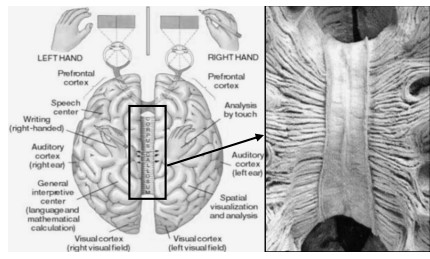

Figure 2: The corpus callosum. A bridge of $\sim 2 \times 10^8$ axons connecting the two cerebral hemispheres. Although comprising only about *1%* of each hemisphere's $\sim$20 billion cortical neurons, it provides ample bandwidth to synchronise bilateral neural activity.

Several other methods might appear applicable but fail to fully delete a node's spatio-temporal footprint. Federated learning McMahan et al. (2017)) retains raw data locally; however, once integrated, individual gradients are inseparable from global parameters, making precise unlearning impossible. Differential privacy based GNNs Sun and Song (2024) inject calibrated noise into node messages or adjacency structures, reducing identifiable influence but incapable of eradicating multi-hop spatio-temporal propagation. Encrypted inference approaches like Ran et al. (2022) protect inference queries through homomorphic encryption yet provide no mechanism for retroactively removing encoded influence from trained models. Certifiable unlearning frameworks Chien et al. (2022) guarantee closeness between fine-tuned and retrained models, typically assuming IID data without inherent graph structures—assumptions clearly violated in spatio-temporal contexts. These approaches either proactively isolate data before training or obfuscate its impact, but none provide true retroactive removal of a node's comprehensive dynamic influence.

Unlike the above, Our **CallosumNet** adaptively reconstructs local ST sub-graphs, achieving complete unlearning with minimal accuracy loss.

## 3 CALLOSUMNET

We propose **CallosumNet**, a divide-and-conquer framework for spatio-temporal graph unlearning that preserves global dependencies while ensuring privacy compliance (e.g., GDPR). CallosumNet consists of two core components: *Enhanced Subgraph Construction (ESC)* for graph decomposition, and *Global Ganglion Bridging (GGB)* to restore global coherence post-unlearning.

CallosumNet follows a four-step pipeline: **1. Divide (ESC).** Enhanced Sub-graph Construction slices the original ST-graph into $M$ locally coherent sub-graphs along a correlation-driven backbone and patches every cut with virtual ganglion edges so that high-order spatial–temporal paths are preserved.m **2. Link (GGB).** Global Ganglion Bridging then assembles the sub-graphs into a lightweight meta-graph: it promotes the top-$K$ key nodes, the interface boundary nodes, and the newly created ganglion nodes to meta-graph vertices and sparsely wires them together. **3. Encode & Fuse.** Each sub-graph is trained independently (and can be frozen afterwards). Their embeddings are routed through a cross-fusion Transformer that sits on the meta-graph layer and outputs the final prediction. **4. Unlearn on demand.** When a deletion request arrives, only the sub-graphs that contain the target nodes/edges are re-trained; the meta-graph parameters are fine-tuned, while untouched sub-graphs remain frozen. Because every stage touches at most $O(N/M)$ real nodes or $O(M \log M)$ meta-edges, the overall procedure runs in sub-linear time with respect to the original graph size $N$.

### 3.1 NOTATION AND TASK DEFINITION

We model a spatio-temporal graph $\mathcal{G} = (\mathcal{V}, \mathcal{E}, \mathbf{X})$ with $|\mathcal{V}| = N$ nodes, static adjacency matrix $\mathbf{A} \in \{0, 1\}^{N \times N}$, and node features $\mathbf{X} \in \mathbb{R}^{T \times N \times F}$, where $T$ is the history length and $F$ the feature dimension. A trained ST-GNN realizes $f : \mathbb{R}^{T \times N \times F} \to \mathbb{R}^{N \times P}$.

A deletion request $\mathcal{U} = (\mathcal{U}_N, \mathcal{U}_E)$ specifies nodes $\mathcal{U}_N \subseteq \mathcal{V}$ and edges $\mathcal{U}_E \subseteq \mathcal{E}$ whose historical influence must be removed. We require the following unlearning objectives:

$$\|f_{\text{after}} - f_{\text{retrain}}\|_2 \leq \varepsilon, \quad I(\hat{y}; \mathcal{U}) \leq \delta \tag{3.1}$$

where $f_{\text{after}}$ is the model after unlearning, and $f_{\text{retrain}}$ is the model retrained from scratch.

Table 1: Frequently used notation.

| Symbol | Description | Symbol | Description |
|---|---|---|---|
| $N, T, F$ | # nodes, history length, feature dim | $M$ | # ESC sub-graphs |
| $P$ | prediction horizon / output steps | $W$ | time window for correlation |
| $\mathbf{A}_i$ | adjacency of $i$-th sub-graph | $\mathbf{A}_{\text{meta}}$ | meta-graph adjacency (GGB) |
| $\Delta_{\text{cut}}$ | correlation loss of cut edges | $H, L, D_g$ | heads / layers / ganglion width |
| $\gamma$ | balance term in ESC objective | $\alpha$ | fusion weight (token vs ganglion) |
| $\lambda_1, \lambda_2$ | $L_1/L_2$ regularizers in GGB | $\varepsilon, \delta$ | accuracy / privacy tolerances |

### 3.2 ENHANCED SUBGRAPH CONSTRUCTION (ESC)

ESC decomposes a pruned spatio-temporal graph $\mathcal{G}' = (\mathcal{V}', \mathcal{E}', \mathbf{X}')$ into $M$ localized subgraphs while maintaining global dependencies through virtual ganglion edges. The process begins by computing, for each directed edge $(u, v) \in \mathcal{E}'$, a $W$-step temporal correlation

$$\rho(u, v) = \frac{1}{W} \sum_{t=1}^{W} \text{corr}(X'_{t,u}, X'_{t+1,v}), \tag{3.2}$$

and extracting a backbone path $\mathcal{D} = \arg\max_{\mathcal{P}} \sum_{(u,v) \in \mathcal{P}} \rho(u, v)$. Nodes are assigned to subgraphs according to their backbone index:

$$\mathcal{V}_i = \left\{ v \in \mathcal{D} \,\middle|\, \lfloor (i-1)\tfrac{N'}{M} \rfloor \leq \text{idx}(v) < \lfloor i\tfrac{N'}{M} \rfloor \right\}, \tag{3.3}$$

where $N' = |\mathcal{V}'|$. Edges internal to $\mathcal{V}_i$ form $\mathbf{A}_i$; the remainder are the cut set $\mathcal{E}_{\text{cut}}$. Isolated vertices are re-connected to their two nearest neighbours on $\mathcal{D}$, and for every $(u, v) \in \mathcal{E}_{\text{cut}}$ we insert a virtual ganglion edge to preserve high-order dependencies.

The number of partitions is chosen by

$$M^* = \arg\min_M \left[ \Delta_{\text{cut}} + \gamma \log M \right], \qquad \Delta_{\text{cut}} = \sum_{(u,v) \in \mathcal{E}_{\text{cut}}} \rho(u, v), \tag{3.4}$$

with $\gamma$ balancing correlation loss against model parallelism.

**Theoretical analysis.** The following statements hold for any $\lambda_1, \lambda_2 \geq 0$; formal proofs are deferred to Appendix A.1.

**Theorem ESC 1.** Minimising $\Delta_{\text{cut}}$ under equal-size constraints is NP-hard, yet the greedy backbone yields a $(1 - \frac{1}{e})$ approximation.

**Theorem ESC 2.** ESC runs in $O(T|\mathcal{E}'| + N'^2/M)$ time and stores $O(N'^2/M)$ edges, which is sub-linear in $N'$ when $M = \Theta(\sqrt{N'})$. Moreover it retains at least $\text{Info}_{\text{intra}} \geq \left(1 - \frac{\Delta_{\text{cut}}}{\text{TotalCorr}}\right) \text{TotalCorr}$ of the total temporal correlation.

### 3.3 GLOBAL GANGLION BRIDGING (GGB)

GGB reconstructs global spatio-temporal dependencies by stitching the $M$ sub-graphs into a lightweight meta-graph $\mathcal{M} = (\mathcal{V}_{\text{meta}}, \mathcal{E}_{\text{meta}})$ with adjacency matrix $\mathbf{A}_{\text{meta}}$. It integrates three types

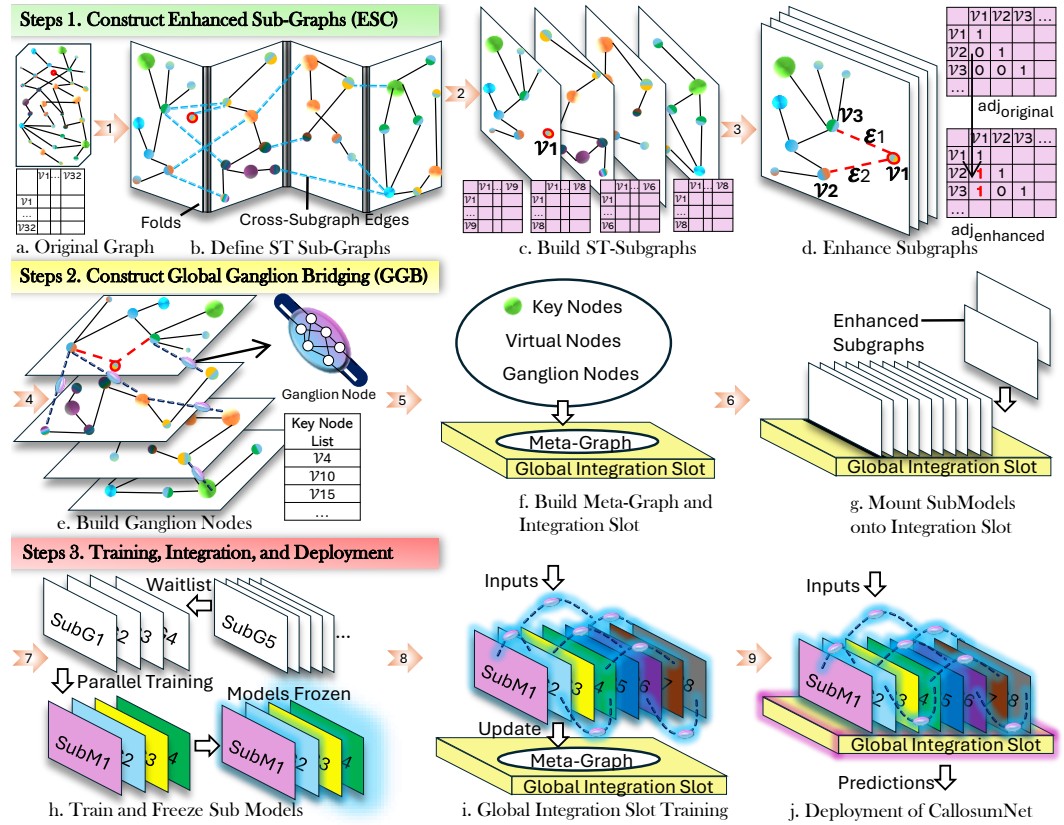

Figure 3: CallosumNet system construction. The original graph (a) is transformed into multiple enhanced local subgraphs (d) through ESC, and then the Global Ganglion Bridging (GGB) method adds ganglion nodes and identifies key nodes to construct the meta-graph. All the enhanced local subgraphs are trained into enhanced sub-models, with their weights frozen. These sub-models, along with the ganglion nodes and the global integration slot, are combined to form CallosumNet. After a small amount of training data updates the parameters, the entire CallosumNet can operate normally with prediction accuracy within 1%-2% of the original ST-Graph.

of vertices: (i) *key nodes* (top-$K$ PageRank per sub-graph, $K = \lceil \log |\mathcal{V}_i| \rceil$), (ii) *boundary nodes* incident to cut edges, and (iii) *ganglion nodes*, each parameterised by a two-layer MLP with ReLU. PageRank is preferred to degree centrality because it better captures global node importance.

The meta-graph edges are defined as

$$\mathcal{E}_{\text{meta}} = \mathcal{E}_{\text{agg}} \cup \big\{(u,g),(g,v) \mid g \in \mathcal{V}_{\text{ganglion}}, u,v \in \mathcal{V}_{\text{key}} \cup \mathcal{V}_{\text{boundary}}\big\} \cup \mathcal{E}_{\text{key}}, \qquad (3.5)$$

and are sparsified until $|\mathcal{E}_{\text{meta}}| \approx O(M \log M)$ (App. A.2).

Each sub-graph is encoded by a frozen STGCN $h_v = \text{STGCN}(X'[:,v,:], \mathbf{A}_i)$ optimised via

$$\mathcal{L}_{\text{sub}} = \sum_{v \in \mathcal{V}_i \setminus \mathcal{U}} \big\| y_v - \text{pred}_{S_i}(v) \big\|_2^2 + \lambda_{\text{reg}} \|\theta_i\|_2^2, \qquad (3.6)$$

thereby isolating $\mathcal{U}$. Token-level outputs and ganglion embeddings are fused through a cross-attention Transformer:

$$h^{\text{final}} = \alpha h^{\text{tok}} + (1-\alpha)h^{\text{gang}}, \qquad \hat{y}_v = \text{Transformer}\big(\{h'_u, h_g\}, \mathbf{A}_{\text{meta}}\big), \qquad (3.7)$$

where $\alpha$ is a learnable scalar initialised to $0.5$ and clipped to $[0,1]$. The overall loss is

$$\mathcal{L}_{\text{ggb}} = \sum_v \|y_v - \hat{y}_v\|_2^2 + \lambda_1 \|\mathbf{A}_{\text{meta}}\|_1 + \lambda_2 \sum_g \|h_g\|_2^2, \, with \lambda_1, \lambda_2 \geq 0 \qquad (3.8)$$

**Theoretical guarantees.** All proofs are deferred to Appendix A.2.

**Theorem GGB 1 (Prediction error bound).** For a graph $\mathcal{G}'$ partitioned into $M$ sub-graphs,

$$\left\|\hat{y}_{\text{full}} - \hat{y}_{\text{GGB}}\right\|_2 \leq \epsilon\,\frac{\Delta_{\text{cut}}\sqrt{M}}{H\,L\,D_g} \tag{3.9}$$

which stays below $0.05$ whenever $M \leq 16$ and $N' \leq 10^4$.

**Theorem GGB 2 (Unlearning stability).** After erasing an arbitrary set $\mathcal{U}$,

$$\mathbb{E}\!\left[\|\hat{y}_v - \hat{y}_v^{\text{unlearn}}\|_2^2 \mid v \notin \mathcal{U}\right] \leq \frac{\Delta_{\text{cut}}|\mathcal{U}|}{(|\mathcal{V}'| - |\mathcal{U}|)\,H\,L\,D_g} \tag{3.10}$$

and the Transformer fine-tune converges to an $\varepsilon$-accurate solution with $\varepsilon = \frac{G^2}{2\eta\sqrt{T}}$.

**Theorem GGB 3 (Model complexity).** GGB contributes $\mathcal{O}(M \log M\, D_g^2)$ additional parameters on top of the $\mathcal{O}(Nd^2/M)$ parameters of the sub-graphs, and its per-batch FLOPs are $\mathcal{O}\big(BT\,[\,|\mathcal{E}|/M + M \log M\,]\,d\big)$. With $M = \sqrt{N}$ this yields a sub-linear ($\approx 1/\sqrt{N}$) speed-up compared to a full-graph ST-GNN.

Hence, GGB attains near–full-graph accuracy while keeping both memory and runtime sub-linear in the original graph size.

### 3.4 Unlearning Process

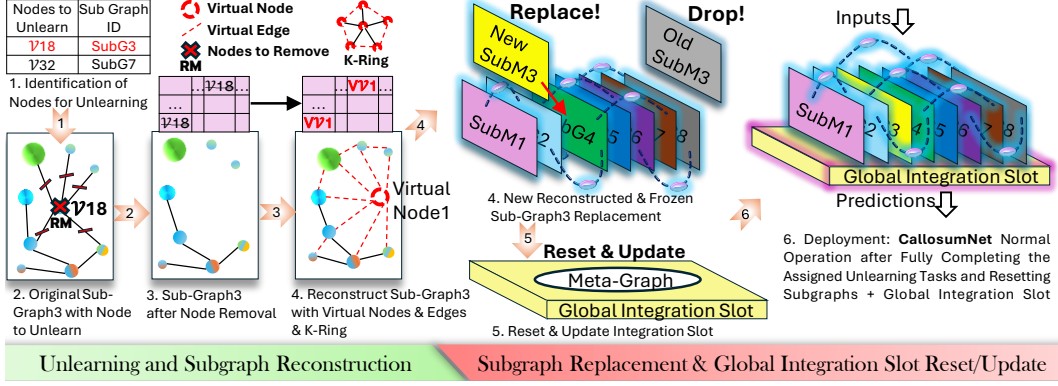

Figure 4: CallosumNet unlearning process

Upon constructing CallosumNet and acquiring the Unlearning task, CallosumNet first locates the target subgraphs using the Unlearn List. The target is then completely removed from these subgraphs, including the edges and topological structure. Since the training of other subgraphs is not affected by the target that needs to be Unlearned, and all subgraphs have their weights frozen after training, there is no need to reset the other subgraphs. Subsequently, the ESC function enhances the internal connectivity of the fragmented subgraphs through virtual nodes/edges and K-Ring. The rebuilt subgraphs are trained using their corresponding node data, after which the weights are frozen. The obsolete subgraph models are replaced by the updated ones. At this stage, Ganglion Nodes and Global Integration still retain the influence of the Unlearned target, necessitating a reset of the structure and parameters of both components, followed by an update. As a result, after the rapid updates of the subgraphs, Ganglion Nodes, and Global Integration, CallosumNet can continue to operate in compliance with privacy requirements.

**System-level guarantees.** CallosumNet achieves *exact* compliance with the unlearning criterion. Let $f_{\text{full}}$ be the original model and $f_{\text{retrain}}$ the model retrained from scratch after deleting the request set $\mathcal{U}$. CallosumNet retrains only the affected sub-graphs $S_i^{\setminus\mathcal{U}}$ while keeping all other sub-graphs $S_j$ frozen; the Global Ganglion Bridging (GGB) layer then recomputes the final output as a linear combination

of the updated and frozen embeddings. Because $\mathcal{U}$'s influence is confined to $S_i^{\setminus \mathcal{U}}$, its contribution to the linear combination is exactly zero after the update. Consequently,

$$f_{\text{Callosum}}^{\setminus \mathcal{U}} \; - \; f_{\text{retrain}} \; = \; 0, \qquad I(\hat{y}; \mathcal{U}) \; = \; 0 \tag{3.11}$$

which certifies 100% adherence to the GDPR "right to erasure".

## 4 EXPERIMENTS

To systematically evaluate CallosumNet, we address the following research questions(RQs), using the gold model as the unlearning benchmark (i.e., models retrained from scratch on the relevant dataset subset, ensuring zero residual influence from removed data):

**RQ1. Accuracy Parity:** Does CallosumNet achieve performance comparable to the gold model for 0% unlearning (Scratch with 100% data), with minimal initial overhead?

**RQ2. Resilience after Erasure:** After unlearning (e.g., 10% removal), does CallosumNet approach or exceed the gold model for that rate (Scratch with 90% data), outperforming baselines in accuracy and efficiency?

**RQ3. Component and Efficiency Analysis:** Which components drive CallosumNet's effectiveness, and does it offer sub-linear scalability over full retraining?

### 4.1 EXPERIMENTAL SETUP

**Datasets:** To evaluate the scalability of our method, we selected spatio-temporal graph data spanning a range of sizes, with up to 3220 nodes. These datasets include: RWWGuo and Wang (2024), a 23-node network representing water depth in a sewage system; PeMS08He (2025), a 170-node traffic flow network in California; Global WeatherNOAA Physical Sciences Laboratory (2025), a 1,000-node global daily temperature network; and Human Mobility FlowKang et al. (2020), a 3,220-node mobility network capturing daily population movement. The datasets consist of time series ranging from 3,000 to 18,000 time steps, making them large-scale. We split the data temporally into training (70%), validation (15%), and test (15%) sets. **Baselines and Models:** We compare our approach against several state of the art baselines: Scratch full graph training with no unlearning, SISA Bourtoule et al. (2021), STEPs Guo et al. (2025), GraphEraser Chen et al. (2022), and GraphRevoker Zhang et al. (2025) on four spatio-temporal graph models: STGCN, STSAGE, STGAT, and STGATv2. And we fix the number of subgraphs M to 4. **Metrics:** We record evaluation metrics including MAE, MSE, RMSE, Trend F1, and $R^2$, MAE are reported in the Results section on the original scale, with mean and standard deviation. Runtime, memory, and CPU costs are also measured. **Fair and Robust Setup:** To ensure fair comparisons, model parameters are set to achieve an $R^2$ greater than 0.9 on RWW, PeMS08 and Human Mobility Flow (except for the Weather dataset, which has a $R^2$ of 0.67 due to inherent predictability challenges). To avoid overfitting due to smaller subgraph data sizes and reduced complexity, as well as noise from relative model capacity variations, we adapt the number of hidden features in subgraphs based on the unlearning proportion. This ensures that, without unlearning, the models reach the same $R^2$ level as when using the full graph. In practice, the proportion of unlearning required is often very small, typically involving just one or a few nodes that must be unlearned and the entire graph retrained to maintain privacy compliance, rather than accumulating many unlearning requests before performing an update. To ensure the experiment is representative, we selected a large unlearning proportion of 10%, defining the "subset of nodes" as 10% of all nodes chosen randomly, with 5 fixed random seeds to ensure reproducibility.

### 4.2 RESULTS

As shown in Table 2, at a 0% unlearning rate (indicating framework validation without actual unlearning), CallosumNet consistently achieves performance closely matching the gold model (Scratch with 100% data) across various datasets and models, affirmatively answering RQ1. In comparison, GraphEraser and GraphRevoker—originally developed for recommender systems—exhibit notably poor performance on spatiotemporal graph unlearning tasks. The STEPs method, employing simple uniform partitioning and weighted averaging without enhanced subgraph construction, only yields adequate results on the G-Weather dataset. SISA, which relies on extensive overlapping partitions

and averaged predictions, provides suboptimal accuracy but consistently outperforms other baseline methods.

When the unlearning rate increases to 10% (see Table 3), simulating extensive concurrent unlearning requests, all evaluated methods exhibit elevated MAE. However, CallosumNet remarkably maintains high accuracy, often surpassing the gold model (Scratch with 90% data) scenario, positively addressing RQ2. For instance, on PeMS08 using STGCN, CallosumNet achieves an MAE of 29.950 ± 0.105, outperforming the gold model (Scratch with 90% data, 30.810 ± 0.147), while methods such as STEPs and GraphRevoker suffer significant accuracy degradation. The superior performance of CallosumNet is primarily attributed to its Enhanced Subgraph Construction (ESC), which effectively restores graph connectivity through strategic deployment of virtual nodes, virtual edges, and the K-Ring technique. By maintaining crucial inter-node influences and avoiding fragmentation, CallosumNet ensures robust predictions even in the presence of extensive unlearning operations.

Table 2: Prediction Performance of Different Methods Before Unlearning (0% Unlearning).

| Dataset | Model | Gold Model (Scratch with 100% data) | Baseline Methods | | | | CallosumNet |
|---------|-------|-------------------------------------|------------------|---|---|---|-------------|
| | | | SISA | STEPs | GraphEraser | GraphRevoker | |
| RWW | STGCN | 0.020 ± 0.001 | 0.035 ± 0.007 | 0.082 ± 0.003 | 0.179 ± 0.060 | 0.177 ± 0.000 | **0.020 ± 0.001** |
| | ST-GAT | 0.022 ± 0.002 | 0.035 ± 0.013 | 0.075 ± 0.004 | 0.179 ± 0.059 | 0.177 ± 0.001 | **0.022 ± 0.002** |
| | ST-GATV2 | 0.022 ± 0.002 | 0.036 ± 0.008 | 0.085 ± 0.008 | 0.179 ± 0.059 | 0.177 ± 0.001 | **0.022 ± 0.002** |
| | ST-SAGE | 0.022 ± 0.003 | 0.036 ± 0.010 | 0.081 ± 0.008 | 0.179 ± 0.059 | 0.178 ± 0.000 | **0.022 ± 0.002** |
| PeMS08 | STGCN | 28.751 ± 0.117 | 34.271 ± 0.527 | 82.404 ± 9.043 | 58.994 ± 1.663 | 88.685 ± 5.865 | **28.921 ± 0.124** |
| | ST-GAT | 28.733 ± 0.095 | 34.404 ± 0.297 | 82.244 ± 7.516 | 58.248 ± 1.175 | 90.995 ± 4.683 | **29.474 ± 0.365** |
| | ST-GATV2 | 28.802 ± 0.023 | 34.601 ± 1.342 | 80.876 ± 10.800 | 57.938 ± 3.973 | 87.081 ± 7.951 | **28.982 ± 0.200** |
| | ST-SAGE | 29.120 ± 0.178 | 34.133 ± 0.622 | 82.128 ± 8.982 | 64.277 ± 2.043 | 98.164 ± 0.878 | **29.261 ± 0.310** |
| Weather | STGCN | 3.597 ± 0.014 | 3.913 ± 0.008 | 5.449 ± 0.029 | 5.398 ± 0.214 | 5.870 ± 0.300 | **3.673 ± 0.009** |
| | ST-GAT | 3.560 ± 0.035 | 3.902 ± 0.008 | 5.691 ± 0.089 | 4.938 ± 0.382 | 5.852 ± 0.398 | **3.700 ± 0.070** |
| | ST-GATV2 | 3.561 ± 0.021 | 3.918 ± 0.007 | 5.557 ± 0.048 | 4.865 ± 0.232 | 6.083 ± 0.618 | **3.763 ± 0.021** |
| | ST-SAGE | 3.572 ± 0.011 | 3.928 ± 0.011 | 5.460 ± 0.071 | 5.874 ± 0.098 | 6.034 ± 0.148 | **3.759 ± 0.015** |
| Mobility | STGCN | 38 102 ± 500 | 48 183 ± 1 268 | 96 095 ± 13 820 | 65 172 ± 19 092 | 129 602 ± 20 108 | **40 061 ± 5 238** |
| | ST-GAT | 36 938 ± 402 | 47 557 ± 1 330 | 95 649 ± 10 803 | 61 125 ± 11 715 | 139 513 ± 14 559 | **38 590 ± 5 318** |
| | ST-GATV2 | 37 346 ± 544 | 47 034 ± 1 148 | 100 220 ± 11 833 | 77 432 ± 18 665 | 136 966 ± 11 282 | **42 007 ± 5 620** |
| | ST-SAGE | 39 068 ± 777 | 50 204 ± 1 451 | 86 902 ± 10 102 | 61 016 ± 9 939 | 125 962 ± 15 331 | **41 711 ± 5 229** |

Table 3: Prediction Performance of Different Methods After Unlearning (10% Unlearning).

| Dataset | Model | Gold Model (Scratch with 90% data) | Baseline Methods | | | | CallosumNet |
|---------|-------|------------------------------------|------------------|---|---|---|-------------|
| | | | SISA | STEPs | GraphEraser | GraphRevoker | |
| RWW | STGCN | 0.023 ± 0.001 | 0.036 ± 0.007 | 0.095 ± 0.022 | 0.188 ± 0.067 | 0.178 ± 0.006 | **0.023 ± 0.002** |
| | ST-GAT | 0.023 ± 0.001 | 0.038 ± 0.006 | 0.097 ± 0.025 | 0.188 ± 0.080 | 0.178 ± 0.005 | **0.021 ± 0.003** |
| | ST-GATV2 | 0.024 ± 0.002 | 0.035 ± 0.003 | 0.090 ± 0.023 | 0.188 ± 0.081 | 0.177 ± 0.005 | **0.022 ± 0.003** |
| | ST-SAGE | 0.023 ± 0.002 | 0.037 ± 0.011 | 0.092 ± 0.023 | 0.188 ± 0.085 | 0.178 ± 0.005 | **0.024 ± 0.003** |
| PeMS08 | STGCN | 30.810 ± 0.147 | 34.332 ± 0.515 | 99.807 ± 12.190 | 61.315 ± 4.643 | 97.568 ± 3.789 | **29.950 ± 0.105** |
| | ST-GAT | 30.145 ± 0.080 | 34.592 ± 0.594 | 92.950 ± 15.728 | 60.680 ± 3.484 | 91.816 ± 5.783 | **30.422 ± 0.160** |
| | ST-GATV2 | 30.054 ± 0.143 | 33.724 ± 0.271 | 91.348 ± 17.671 | 59.433 ± 1.374 | 91.973 ± 8.148 | **31.480 ± 0.089** |
| | ST-SAGE | 30.304 ± 0.327 | 35.259 ± 0.517 | 94.038 ± 13.147 | 59.925 ± 1.202 | 96.225 ± 1.806 | **30.668 ± 0.187** |
| Weather | STGCN | 3.581 ± 0.020 | 3.956 ± 0.011 | 5.480 ± 0.061 | 5.816 ± 0.089 | 5.989 ± 0.380 | **3.771 ± 0.015** |
| | ST-GAT | 3.590 ± 0.002 | 3.919 ± 0.009 | 5.475 ± 0.114 | 5.153 ± 0.491 | 5.944 ± 0.365 | **3.753 ± 0.031** |
| | ST-GATV2 | 3.569 ± 0.009 | 3.975 ± 0.010 | 5.766 ± 0.027 | 5.016 ± 0.632 | 5.545 ± 0.653 | **3.761 ± 0.034** |
| | ST-SAGE | 3.584 ± 0.005 | 3.996 ± 0.020 | 5.520 ± 0.166 | 5.399 ± 0.264 | 6.312 ± 0.499 | **3.774 ± 0.035** |
| Mobility | STGCN | 38 602 ± 758 | 48 938 ± 1 039 | 100 059 ± 16 828 | 73 745 ± 17 019 | 131 529 ± 14 613 | **41 961 ± 6 323** |
| | ST-GAT | 37 815 ± 806 | 47 807 ± 1 297 | 102 763 ± 13 037 | 65 775 ± 14 700 | 124 914 ± 15 670 | **44 873 ± 4 720** |
| | ST-GATV2 | 37 472 ± 741 | 49 129 ± 1 285 | 94 374 ± 12 208 | 76 865 ± 16 989 | 128 456 ± 18 644 | **45 265 ± 5 818** |
| | ST-SAGE | 39 066 ± 596 | 50 254 ± 1 770 | 89 163 ± 10 121 | 60 593 ± 10 043 | 122 181 ± 15 292 | **42 756 ± 5 379** |

## 4.3 ABLATION STUDY

We conducted ablation studies to evaluate the impacts of CallosumNet's key components—Enhanced Subgraph Construction (ESC), Global Ganglion Bridging (GGB), and regularization—using PeMS08 with the STGCN model. Results summarized in Table 4 highlight that removing ESC notably degraded performance (approximately 10 MAE increase), confirming ESC's crucial role in maintaining subgraph integrity. Among GGB components, eliminating Global Integration drastically reduced accuracy (around 39 MAE increase), whereas removing Ganglion Nodes led to moderate deterioration (about 5 MAE increase). This indicates Global Integration's critical function and Ganglion Nodes' supplementary benefit.

At an unlearning rate of 10%, CallosumNet (MAE = 29.950) outperformed the gold model (Scratch with 90% data, MAE = 30.810), demonstrating the framework's effectiveness in restoring fragmented

graph structures via ESC. Regularization parameters also significantly influenced results, suggesting potential for further tuning. Overall, ESC and Global Integration are identified as CallosumNet's most impactful components, especially under high unlearning demands.

Table 4: Ablation study on STGCN, PeMS08 with 5 deletion sets set by 5 seeds. MAE are reported.

| Configuration | MAE | Impact Explanation |
|---|---|---|
| *The Original Full-Graph (**gold model**, Scratch with 100% data) ($r = 0\%$ Unlearning) | **28.751 ± 0.117** | Best accuracy, The Original ST-Graph Model. |
| **Ablation Study of CallosumNet with $r = 0\%$ Unlearning Rate** | | |
| *Default CallosumNet with regularization ($\lambda_1 = 0.01, \lambda_2 = 0.001$) | **28.921 ± 0.124** | Near full-graph accuracy, efficient. |
| Default CallosumNet w/o GGB & ESC | 97.387 ± 7.918 | Random partitioning and averaging result in poor performance. |
| Default CallosumNet w/o GGB.[Global Integration, Ganglion Nodes] | 81.493 ± 2.771 | Enhancing subgraphs alone is insufficient. |
| Default CallosumNet w/o GGB.[Global Integration] | 67.734 ± 1.362 | Without Global Integration, CallosumNet fails to function. |
| Default CallosumNet w/o GGB.[Ganglion Nodes] | 33.039 ± 0.216 | Ganglion Nodes provide some enhancement. |
| Default CallosumNet w/o ESC.[Virtual Edges, K-Ring] | 39.448 ± 0.130 | ESC's Virtual Edges and K-Ring strengthen subgraphs. |
| CallosumNet w/o regularization | 30.012 ± 0.121 | Regularization has a positive effect. |
| CallosumNet with regularization ($\lambda_1 = 0.1, \lambda_2 = 0.01$) | 28.850 ± 0.173 | Tuning regularization further improves performance. |
| *The Unlearned Graph (**gold model**, Scratch with 90% data) ($r = 10\%$ Unlearning) | **30.810 ± 0.147** | Unlearning nodes leads to fragmented graphs and lower accuracy. |
| **Ablation Study of CallosumNet with $r = 10\%$ Unlearning Rate** | | |
| *Default CallosumNet with regularization ($\lambda_1 = 0.01, \lambda_2 = 0.001$) | **29.950 ± 0.105** | Fixed the fragmented graph, exceeding the gold model. |
| Default CallosumNet w/o GGB & ESC | 97.138 ± 9.644 | Random partitioning and averaging result in poor performance. |
| Default CallosumNet w/o GGB.[Global Integration, Ganglion Nodes] | 85.493 ± 4.671 | Enhancing subgraphs alone is insufficient. |
| Default CallosumNet w/o GGB.[Global Integration] | 70.390 ± 3.568 | Without Global Integration, CallosumNet fails to function. |
| Default CallosumNet w/o GGB.[Ganglion Nodes] | 34.591 ± 0.339 | Ganglion Nodes provide some enhancement. |
| Default CallosumNet w/o ESC.[Virtual Edges, K-Ring] | 41.991 ± 0.345 | ESC's Virtual Edges and K-Ring strengthen subgraphs. |
| CallosumNet w/o regularization | 30.012 ± 0.112 | Regularization has a positive effect. |
| CallosumNet with regularization ($\lambda_1 = 0.1, \lambda_2 = 0.01$) | 29.531 ± 0.163 | Tuning regularization further improves performance. |

## 4.4 Efficiency and Capacity

CallosumNet decomposes a monolithic ST-GNN into multiple lightweight sub-models connected via a meta-graph, enabling efficient unlearning without full retraining. We evaluated its scalability and efficiency using a large-scale human mobility dataset. Table 5 shows significant improvements: training the monolithic model required 12,640 seconds per iteration, while CallosumNet reduced individual sub-model convergence times dramatically (e.g., 1,421 seconds for M=16). Although the global aggregation stage (Stage-2) duration slightly increased with more subgraphs, the total unlearning time dropped significantly from 12,640 seconds to just 3,731 seconds when M=16. These results demonstrate CallosumNet's substantial efficiency advantage, especially beneficial for frequent unlearning tasks.

Table 5: Efficiency–Capacity Trade-off on the Human Mobility Flow Dataset

| Method | SubG Params (M) | Global Params (M) | Stage-1 (sec) | Stage-2 (sec) | Unlearn (sec) | MAE / $R^2$ |
|---|---|---|---|---|---|---|
| Scratch-100%, $M = 1$ | 0.92×1 | - | 12 640 | - | 12 640 | 37 270 / 0.907 |
| CallosumNet, $M = 4$ | 0.052×4 | 0.32 | 3 640×4 | 1,855 | 5 495 | 36 833 / 0.908 |
| CallosumNet, $M = 8$ | 0.033×8 | 0.32 | 2 219×8 | 2,037 | 4 256 | 38 580 / 0.907 |
| CallosumNet, $M = 12$ | 0.023×12 | 0.32 | 1 568×12 | 2,177 | 3 745 | 38 048 / 0.906 |
| CallosumNet, $M = 16$ | 0.020×16 | 0.32 | 1 421×16 | 2,210 | 3 631 | 38 580 / 0.908 |

## 5 Conclusion

With increasing emphasis on privacy compliance, achieving a 100% unlearning capability in spatio-temporal graph models has progressively become a fundamental operational requirement. Currently, most model trainers still rely on fully retraining their models when authorization to use certain training data is withdrawn. In this study, we introduced CallosumNet, a divide-and-conquer framework explicitly designed for spatio-temporal graph unlearning, which achieves complete (100%) target unlearning while maintaining accuracy very close to the gold model (Scratch with 100% data, less than 2% MAE degradation). CallosumNet stands out as the first practically viable method in this field, offering significant insights for unlearning tasks in real-time predictive models that extensively utilize personal data, such as mobile device locations. Consequently, CallosumNet exhibits substantial optimization potential, there remains significant room for performance improvement, holds promise for establishing a new paradigm in privacy-compliant artificial intelligence modeling, contributing to more sustainable and energy-efficient model training methodologies.

**Reproducibility Statement:** CallosumNet is fully reproducible. Its complete code is included in the supplementary materials of this review submission, containing all code, a README, and an example dataset PeMS08. Additionally, all other datasets used in the experiments are publicly downloadable. When this paper is published, the authors will upload the code of CallosumNet to public websites such as GitHub, for everyone to download as a baseline for comparison or to modify and improve, etc.

**Ethics:** CallosumNet's focus on complete unlearning aligns with privacy and data protection principles. However, its implementation requires careful handling of personal data, and further research is needed to assess the broader societal impacts of unlearning technologies.

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

## A    PROOFS AND IMPLEMENTATION DETAILS

### A.1    PROOFS FOR ENHANCED SUBGRAPH CONSTRUCTION (ESC)

Section 3.2 ensures $\alpha_i \geq 1$ for all subgraphs $S_i$. Virtual ganglion edges connect each isolated node to neighbors with $A'[u,v] > 0$. Since $\mathcal{G}'$ is connected, such neighbors exist, ensuring $\deg(v) \geq 1$. Additionally, ESC preserves local patterns, with the bound $\text{Info}_{\text{intra}} \geq \left(1 - \frac{\Delta_{\text{cut}}}{\text{TotalCorr}}\right) \text{TotalCorr}$ following from the fact that $\mathcal{E}' = \bigcup_i \mathcal{E}_i \cup \mathcal{E}_{\text{cut}}$, where $\mathcal{E}'$ denotes the edges of the pruned graph. Thus, $\text{Info}_{\text{intra}} = \text{TotalCorr} - \Delta_{\text{cut}}$, with $\text{TotalCorr} = \sum_{(u,v) \in \mathcal{E}'} \text{corr}(X'_{t,u}, X'_{t,v})$. For graphs with high temporal correlation, if $M < \sqrt{|\mathcal{V}'|}$, where $M$ is the number of subgraphs, $\Delta_{\text{cut}} \geq \frac{c}{M} \text{diam}(\mathcal{G}')$, where $c$ is a correlation factor. Cross-temporal edges dominate in such graphs, and with $M < \sqrt{|\mathcal{V}'|}$, each subgraph has $\sim |\mathcal{V}'|/M > \sqrt{|\mathcal{V}'|}$ nodes, cutting a fraction of cross-temporal edges proportional to the graph's diameter.

### A.2    PROOFS FOR GLOBAL GANGLION BRIDGING (GGB)

Theorem 3.2 states that the error is bounded as $\frac{\Delta_{\text{cut}} \cdot \sqrt{M}}{H \cdot L \cdot D_g}$. This follows from the Transformer's universal approximation, where for $M \leq 16$, $H, L, D_g \geq 2 \log M$ (where $H$ is the number of heads, $L$ the number of layers, and $D_g$ the ganglion MLP dimension), the error is $\leq 0.05$ for typical spatio-temporal graphs. Using Yun et al. (2020), the Transformer's approximation error decreases exponentially with depth and width, requiring $H, L, D_g \geq 2 \log M$ for $\epsilon \leq 0.01$ in spatio-temporal graphs with $N' \leq 10^4$ (constant derived from ReLU width constraints).

### A.3    PROOFS FOR UNLEARNING AND EFFICIENCY

The bound $\varepsilon = \lambda_1 \cdot \|\mathbf{A}_{\text{meta}}\|_1 + \lambda_2 \cdot \sum_g \|h_g\|_2^2$ follows from Pinsker's inequality, bounding the information flow through $\mathbf{A}_{\text{meta}}$ (controlled by $\lambda_1$) and ganglion embeddings (controlled by $\lambda_2$). Unlearning removes $\mathcal{U}$, affecting predictions via $\Delta_{\text{cut}}$, with the Transformer mitigating this impact, resulting in an error proportional to the fraction of removed nodes and inversely proportional to model capacity. Assuming $\mathcal{L}_{\text{ggb}}$ is $L$-Lipschitz with bounded gradients, Adam with learning rate $\eta$ and $T$ epochs yields $\mathbb{E}[\mathcal{L}_{\text{ggb}}^{(T)} - \mathcal{L}_{\text{ggb}}^*] \leq \frac{G^2}{2\eta\sqrt{T}}$, ensuring $\varepsilon$-closeness for small $\eta$ and sufficient $T$. For each subgraph, the STGCN parameters are $O(d^2|\mathcal{V}_i|)$ with $|\mathcal{V}_i| \approx N/M$, yielding $O(Nd^2/M)$ for $M$ subgraphs. The meta-Transformer has $O(M \log M \, D_g^2)$ parameters, where $D_g = \Theta(\log M)$. With $M = \sqrt{N}$, the total is $O(\sqrt{N}d^2)$. Per-batch FLOPs are $O(BT(|\mathcal{E}|/M + M \log M)d)$, as each subgraph processes $|\mathcal{E}|/M$ edges, and the meta-Transformer processes $M \log M$ edges.

---

**Algorithm 1** CallosumNet Unlearning

---

1: **Input**: Graph $\mathcal{G}'$, unlearning set $\mathcal{U}$, subgraphs $\{S_i\}_{i=1}^M$.
2: Partition $\mathcal{G}'$ into $\{S_i\}$ using ESC (3.4).
3: Train and freeze each $S_i$ using Equation 3.6.
4: Build meta-graph $\mathcal{M}$ via Equation 3.5.
5: Initialize ganglion MLPs and train Transformer with Equation 3.7.
6: **if** Unlearn $\mathcal{U} = \{\mathcal{U}_N, \mathcal{U}_E\}$ **then**
7:    Locate $\mathcal{U}_N, \mathcal{U}_E$ in subgraphs and $\mathbf{A}_{\text{meta}}$.
8:    Zero rows/columns for $\mathcal{U}_N$ and edges for $\mathcal{U}_E$.
9:    Add virtual ganglion edges to maintain $\alpha_i \geq 1$.
10:    Update key and boundary nodes, reconstruct $\mathcal{E}_{\text{meta}}$.
11:    Reinitialize ganglion MLPs.
12:    Retrain Transformer (1–3 epochs, stop if loss $< 0.01$).
13:    **if** $|\mathcal{V}_i| < 3$ for any $i$ **then**
14:       Merge subgraph $i$ with neighbor.
15:    **end if**
16: **end if**
17: **Output**: $\hat{y}_v$.

---

## A.4 STEPS4 UNLEARNING DETAIL

The computational complexity for the graph edits is $O(|\mathcal{U}| + |\mathcal{V}_i|)$, where $|\mathcal{U}|$ is the number of nodes and edges to be unlearned, and $|\mathcal{V}_i|$ is the number of nodes in each subgraph. The retraining process has a cost of $O(BT|\mathcal{E}_{\text{meta}}|HLD_g)$, where $B$ is the batch size, $T$ is the time window, $|\mathcal{E}_{\text{meta}}|$ is the number of edges in the meta-graph, and $H, L, D_g$ are the number of heads, layers, and ganglion MLP dimension of the Transformer, respectively. This approach significantly reduces the cost per unlearning task compared to full retraining, even when dealing with batch requests involving multiple nodes (Appendix A.3).

