# OpenReview forum: "Spatio-Temporal Graph Unlearning"
_ICLR.cc/2026/Conference — ICLR 2026 Conference Withdrawn Submission_

### Official Review · Reviewer_Mjfe · 2025-10-26

**Soundness:** 2
**Presentation:** 2
**Contribution:** 2
**Rating:** 4
**Confidence:** 4

**Summary:**

This paper addresses the requirement for the "right to be forgotten" in spatiotemporal graph data under privacy regulations by proposing the CallosumNet framework. Inspired by the structure of the corpus callosum, the framework employs a "divide and conquer" strategy to achieve complete erasure of individual node information. Experiments demonstrate that it incurs minimal performance loss and significantly outperforms existing methods.

**Strengths:**

1. This paper focuses on a unique problem.
2. Experiments demonstrate the effectiveness of the model.

**Weaknesses:**

- One of the biggest flaws of the paper is that I cannot see the necessity of unlearning in spatio-temporal forecasting. The authors mention privacy issues, but in reality, most spatio-temporal datasets are open-source and do not involve personal information. The statement, "As a result, ensuring compliance with these privacy requirements often requires retraining the entire spatio-temporal graph model to preserve privacy for individual nodes, a process that, while essential, introduces additional computational demands," raises questions. Why is retraining needed? What are the specific necessity and application scenarios for individually protecting the privacy of a particular node?

- The authors emphasize interpretability in the introduction, but the experimental section does not provide significant or clear demonstrations aligned with this claim.

- The reasons why existing unlearning algorithms fail need further explanation.

- Section 4.1 is suggested to be divided into subsections for better structure.

- What is the principle for selecting baselines? More advanced models such as STID, D2STGNN, BIST, and BIGST should be included for comparison. Furthermore, the unlearning methods used as baselines require further justification and explanation.

- The literature review is insufficient. With only 18 related works cited, it is difficult to convince reviewers that this adequately summarizes existing work. Consequently, I cannot clearly identify the specific shortcomings of existing technologies that this paper aims to address.

- The authors should include an ablation study variant where the inputs (and edges, if necessary) for the removed nodes are set to zero. This scenario transforms the problem into a missing value imputation problem. Furthermore, I believe some powerful Transformer-based spatio-temporal graph neural networks might handle this effectively.

- The formatting of equations and symbols seems odd. The template likely requires numbering like (1), not (3.1).

- The problem definition is unclear. Are the two coefficients in Section 3.1 hyperparameters? Is the objective of unlearning to measure the parameter distance between two models? How is this objective reflected in the experiments? Also, what does 'I' represent?

- How are the nodes selected for removal in the experiments? Was cross-validation performed to check the robustness of the method to different missing patterns?

- The results are mostly presented in tables. It is recommended to include more figures for illustration and clarity.

- The methodology section is difficult to understand. The authors do not provide sufficient background for each section, and many variables lack intuitive explanations, requiring significant effort to comprehend the proposed framework.

**Questions:**

No additional questions.

---

### Official Review · Reviewer_yf4o · 2025-10-31

**Soundness:** 2
**Presentation:** 3
**Contribution:** 2
**Rating:** 2
**Confidence:** 4

**Summary:**

This paper introduces CallosumNet, a divide-and-conquer framework for spatio-temporal graph unlearning—a largely underexplored but increasingly critical problem due to privacy regulations like GDPR and CCPA. The authors propose a biologically inspired architecture (based on the corpus callosum) that decomposes a global spatio-temporal graph into localized subgraphs using Enhanced Subgraph Construction (ESC) and then reconstructs global dependencies via a lightweight meta-graph using Global Ganglion Bridging (GGB). This design enables exact unlearning of nodes/edges with minimal retraining cost, while preserving predictive accuracy. Empirical results on four real-world datasets show that CallosumNet achieves 100% unlearning with only 1–2% MAE degradation, significantly outperforming existing baselines.

**Strengths:**

- Originality: This is the first principled framework for exact unlearning in spatio-temporal graphs, a domain where influence propagates across both space and time. The biological inspiration (corpus callosum) is novel and well-justified. The idea of virtual ganglion edges and meta-graph fusion is creative and technically sound.

- Technical Quality: The paper is rigorous in both design and theory. The authors provide formal guarantees for unlearning exactness, prediction error bounds, and model complexity. The ESC and GGB modules are well-defined, and the algorithmic pipeline is clear and reproducible.

- Clarity: Despite the complexity of the task, the paper is well-structured and clearly written. The motivation, method, and evaluation are logically organized. Figures (e.g., Fig. 1, Fig. 3, Fig. 4) effectively illustrate the unlearning process and system architecture.

- Significance: The work addresses a real-world, urgent problem—how to efficiently and completely remove user data from trained spatio-temporal models without full retraining. This is especially relevant for mobile services, smart cities, and health monitoring systems, where GDPR compliance is mandatory. CallosumNet offers a practical and scalable solution.

**Weaknesses:**

- Limited Theoretical Analysis of Approximation Error: While the paper provides unlearning exactness, it lacks a fine-grained analysis of how approximations in ESC and GGB affect long-term prediction stability. For example, how does the meta-graph fusion affect error propagation over time? A perturbation analysis or error bound under cumulative unlearning would strengthen the contribution.

- Scalability to Larger Graphs: Although the method is tested on graphs with up to 3,220 nodes, urban-scale graphs (e.g., 10K+ nodes) are not considered. The meta-graph construction and ganglion node selection may become bottlenecks in such cases. A complexity scaling analysis with respect to graph size, subgraph number M, and time steps T would be helpful.

- Assumption of Frozen Subgraphs: The assumption that subgraphs can be frozen after initial training may not hold in non-stationary environments (e.g., concept drift in traffic or mobility). In such cases, subgraph retraining may be necessary, which could reduce the efficiency gain. A discussion or experiment on dynamic graphs would be valuable.

- Limited Baseline Diversity: While the paper compares against 5 baselines, no comparison with recent unlearning methods from graph rewiring, influence cancellation, or gradient-based unlearning is provided. Also, no comparison with differential privacy or certified removal methods is made, which could help contextualize the exact unlearning claim.

- Ethical and Societal Implications: Although the paper includes an ethics statement, it is brief and generic. Given the privacy-sensitive nature of the application, a deeper discussion on potential misuse (e.g., unlearning to hide malicious behavior) or failure modes (e.g., incomplete unlearning due to meta-graph leakage) would be appropriate.

**Questions:**

- Approximation vs. Exactness Trade-off: How does CallosumNet behave under repeated unlearning requests? Does the meta-graph fusion introduce approximation drift over time? Have you tested cumulative unlearning scenarios?

- Dynamic Graphs: How does the method perform on non-stationary graphs where subgraphs may need retraining due to concept drift? Does freezing subgraphs still hold in such cases?

- Scalability: What is the largest graph size (in nodes and edges) that CallosumNet can handle efficiently? How does the meta-graph complexity scale with M and N? Have you tested M > 16?

- Comparison with Certified Removal: How does CallosumNet compare with certified unlearning methods (e.g., Newton-type updates, influence functions, gradient residual cancellation) in terms of accuracy, efficiency, and privacy guarantees?

- Adversarial Robustness: Is CallosumNet robust to adversarial unlearning requests (e.g., an attacker trying to hide influence by strategically requesting unlearning)? Can the meta-graph be poisoned or manipulated?

---

### Official Review · Reviewer_BpCa · 2025-11-01

**Soundness:** 3
**Presentation:** 3
**Contribution:** 3
**Rating:** 6
**Confidence:** 2

**Summary:**

This paper addresses the problem of unlearning in spatio-temporal graph models under privacy regulations such as GDPR. The authors argue that existing unlearning methods designed for static graphs are inadequate for dynamic spatio-temporal data due to global diffusion of node influence. To address this, they propose CallosumNet, a divide-and-conquer framework inspired by the corpus callosum structure in the brain. CallosumNet introduces two components: Enhanced Subgraph Construction (ESC), which partitions the graph into locally coherent subgraphs, and Global Ganglion Bridging (GGB), which restores global dependencies through a meta-graph and Transformer-based fusion. Experiments on four datasets (RWW, PeMS08, Global Weather, Human Mobility) show that CallosumNet achieves comparable accuracy to full retraining with significant computational savings.

**Strengths:**

The overall framework is clearly described, and the motivation is easy to follow. The authors conduct comprehensive experiments across multiple datasets and baselines, showing that their method achieves better performance than existing approaches while preserving accuracy close to the gold model. The results demonstrate consistent improvements, and the ablation study helps clarify the role of each component.

**Weaknesses:**

The related work section, while broad, lacks a deep comparison with recent developments in efficient retraining or federated unlearning. Experimental baselines are not comprehensive; several more recent or specialized spatio-temporal unlearning methods are omitted. The analysis of results remains mostly descriptive without strong theoretical or empirical evidence for the claims and there is no practical validation of privacy compliance.

**Questions:**

How does the proposed method verify or certify complete unlearning beyond theoretical claims? For instance, does the framework empirically confirm the removal of latent influence on predictions through influence-function or gradient-based analysis?

---

### Note · Authors · 2025-11-12

**Comment:**

We sincerely thank the area chair and reviewers for their valuable and insightful feedback. Spatial-Temporal Graph Unlearning is an increasingly important and challenging research direction. We will continue improving both the theoretical and practical aspects of our work and expand the experimental scope. Thank you again for your thoughtful comments and efforts.

**Withdrawal Confirmation:**

I have read and agree with the venue's withdrawal policy on behalf of myself and my co-authors.